# Rethinking supervised learning: insights from biological learning and from calling it by its name

## Abstract

The renaissance of artificial neural networks was catalysed by the success of classification models, tagged by the community with the broader term *supervised learning*. The extraordinary results gave rise to a hype loaded with ambitious promises and overstatements. Soon the community realised that the success owed much to the availability of thousands of labelled examples and *supervised learning* went, for many, from glory to shame: Some criticised deep learning as a whole and others proclaimed that the way forward had to be "alternatives" to supervised learning: *predictive*, *unsupervised*, *semi-supervised* and, more recently, *self-supervised learning*. However, these seem all brand names, rather than actual categories of a theoretically grounded taxonomy. Moreover, the call to banish supervised learning was motivated by the questionable claim that humans learn with little or no supervision and are capable of robust out-of-distribution generalisation. Here, we review insights about learning and supervision in nature, revisit the notion that learning and generalization are *not* possible without supervision or inductive biases and argue that we will make better progress if we just call it by its name.

## 1 Introduction

The re-emergence of deep learning during the last decade due to the noteworthy achievements of artificial neural networks (ANN) built up a sort of philosophy that nearly anything could be automatically learnt from data without human intervention, in contrast to the previous approaches:

> [hand designing good feature extractors, engineering skill and domain expertise] can all be avoided if good features can be learned automatically using a general-purpose learning procedure. This is the key advantage of deep learning (LeCun et al., 2015).

Read in hindsight, this claim was clearly an overstatement. The success of deep learning has required iterative hand design of network architectures and techniques that demanded collective, high engineering skill and large doses of interdisciplinary domain expertise. Furthermore, deep learning owes much to the immense computational power poured into training artificial networks (Amodei & Hernandez, 2018; Schwartz et al., 2019) and to the human effort of manually collecting and labelling thousands of images and other data modalities (Russakovsky et al., 2015; Cao et al., 2018). However, the gist of the claim has permeated machine learning research and is pervasive up to these days.

The realisation that the success of deep learning was largely due to the availability of huge labelled data sets prompted various reactions: some authors strongly questioned the usefulness of the algorithms (Marcus, 2018); some delved into the question of whether neural networks generalise beyond or simply memorise the training examples (Zhang et al., 2017; Arpit et al., 2017); and some proposed

new research horizons that can be overly ambitious and potentially misleading: "learning a class from a single labelled example", based on the statement that "humans learn new concepts with very little supervision, [but] the standard supervised deep learning paradigm does not offer a satisfactory solution for learning new concepts rapidly from little data" (Vinyals et al., 2016). As a consequence, multiple research programmes, with various brand names, followed up with the aim of minimising or removing the need for "supervision" to train neural networks: *few-shot*, *one-shot*, *zero-shot*, *predictive*, *unsupervised*, *semi-supervised* and *self-supervised* learning are only a few popular examples.

Exploring alternatives to *classification* and improving the efficiency of learning algorithms should indeed be a priority of machine learning research. As a matter of fact, related approaches have been subject of study since long before the explosion of deep learning (Hinton & Sejnowski, 1999; Chapelle et al., 2006). However, the current publication and discussion trends in the field denote overambitious promises that are in part based on misconceptions and overstatements about biological learning, and amplified by overselling nomenclature. While much of the research output derived from these programmes does provide us with useful techniques and insight, it leaves behind a landscape of confusing terminology and tangled research directions that are hard to navigate and lead many astray.

In this paper, we reflect upon fundamental concepts in machine learning such as supervision, inductive biases and generalisation, which in spite of resting on theoretical grounds, are at the core of misconceptions and overstatements about deep learning commonly seen in the literature. First, we review aspects from biological learning, and compare them to the traits often (mis)attributed to human learning and generalisation in the machine learning literature (Section 2). Second, we revisit insights from classical statistical learning and critically review the terminology and current trends in deep learning research (Section 3). Altogether, we aim at tempering certain claims and promises of deep learning, helping mitigate the confusion over the terminology and suggesting desirable—in our opinion—directions and changes in machine learning research.

## 2   Supervision in biological learning

The link between artificial intelligence—specifically artificial neural networks (Rosenblatt, 1958; Fukushima & Miyake, 1982)—and biological learning systems is intrinsic to the field, as one long-term goal of artificial intelligence is to mirror the capabilities of human intelligence. However, these capabilities are, in our view, often overestimated. One example is the argument that intelligence in nature evolves without supervision and is capable of robust out-of-distribution generalisation. In particular, it is often claimed that humans and other animals learn to visually categorise objects with little or no supervision from a few examples (Vinyals et al., 2016; Marcus, 2018; Morgenstern et al., 2019). In what follows, we will discuss three aspects of biological learning to argue against this view, so as to gain insights that better inform our progress in machine learning: first, we will discuss how generalisation requires exposure to relevant *training* data; second, we will review the variety of supervised signals that the brain has access to; third, we will comment on the role of evolution and brain development.

### 2.1   Generalisation requires exposure to relevant training data

In the argument that machine learning models should generalise from a few examples, there seems to be a promise or aspiration that future better methods will be able to perform robust visual object categorisation—for instance—among many object classes after being trained on one or a few examples per class. While a primary objective is to develop techniques that efficiently extract the maximum possible information from the available examples, we should also remind ourselves that no machine learning algorithm can robustly learn anything that cannot be inferred from the data it has been trained on. Although this may seem to contradict certain current trends and statements in the literature, we should also bear in mind that learning in nature is not different.

First, the amount of data that animals and humans in particular are exposed to is often underestimated. A biological brain continuously receives, processes and integrates multimodal inputs from various sensors—images (light), sound, smell, etc. Humans do not learn to recognise objects by looking at photos from ImageNet, but are rather exposed to a continuous flow of visual stimuli with slow changes of the viewing angle and lighting conditions. Furthermore, the stimuli are coherent across modalities, we are allowed to interact with the objects and we even receive multiple supervision signals, as we discuss later.

The exposure to so much training data makes the human visual system remarkably robust, but still its capabilities are optimised for the tasks it needs to perform and largely determined by the *training data distribution*—and years of evolution, as we will discuss below. For instance, a well-studied property of human vision is that our face recognition ability is severely impaired if faces are presented upside down (Yin, 1969; Valentine, 1988). Setting aside the specific complexity of face processing in the brain, a compelling explanation for this impairment is that we are simply not used to seeing and recognising inverted faces. More generally, while human perception of objects is largely invariant under certain conditions (Biederman & Bar, 1999), object recognition is sensitive to changes in view angle (Tarr et al., 1998), especially when we see objects from unfamiliar viewpoints (Edelman & Bülthoff, 1992; Bülthoff & Newell, 2006; Milivojevic, 2012).

Furthermore, although better than the *one-shot* or *few-shot* generalisation of current ANNs, humans also have limited ability to recognise truly novel classes (Morgenstern et al., 2019). Interestingly, experiments with certain novel classes of objects known as *Greebles* showed that, with sufficient training, humans can acquire expertise in recognising new objects from different viewpoints, even making use of an area of the brain—the fusiform face area—that typically responds strongly with face stimuli (Gauthier et al., 1999). This provides evidence that recognition from multiple viewpoints is possible but only developed after exposure to similar conditions, that is relevant data. This is reminiscent of the effectiveness of data augmentation in deep learning, compared to more naïve regularisation methods (Hernández-García & König, 2018).

The need for exposure to relevant stimuli challenges the notion that humans are capable of strong out-of-distribution generalisation. Rather, it seems that the *transfer learning* capabilities of humans are limited to relatively small changes in the data distribution. A compelling example is our difficulty to learn new languages: someone who natively speaks or has learnt Spanish will be able to transfer a significant amount of knowledge if they are to learn Italian, due to the overlap in the data distribution, but they will have very little to transfer for learning Kanien'kéha or Mandarin.

## 2.2 Supervised signals for the brain

Another commonly found argument has it that children—animals in general—learn robust object recognition without supervision: "a child can generalize the concept of 'giraffe' from a single picture in a book" (Vinyals et al., 2016). First of all, we should mention the role of evolution (expanded in Section 2.3), which can be interpreted as a pre-trained model, optimised through millions of years of data with natural selection as a supervisory signal (Zador, 2019). Second, there is abundant evidence to argue against the very claim that children—and adults—learn in fully unsupervised fashion.

Obviously, the kind of supervision that humans make use of is not that of classification algorithms— we do not see a class label on top of every object we look at. However, we receive supervision from multiple sources. Even though not for every visual stimulus, children do frequently receive information about the object classes they see. For instance, parents would point at objects and name them, then we learn how to read, and generally play a crucial role as teachers in language development (Kuhl, 2007). Non-human animals such as zebra finches learning to sing have also been found to rely on feedback (supervision) from the female adult and not just imitation Carouso-Peck & Goldstein (2019). Furthermore, humans usually follow guided hierarchical learning: children do not directly learn to tell apart breeds of dogs, but rather start with umbrella terms and then progressively learn down the class hierarchy (Bornstein & Arterberry, 2010; Spriet et al., 2021). Gopnik (2021) has asserted that "we learn more from other people than we do from any other source" and Hasson et al. (2020) mention other examples of supervision from *social cues*, that is from other humans, such as learning to recognise individual faces, produce grammatical sentences, read and write; as well as from embodiment and action, such as learning to balance the body while walking or grasping objects. In all these actions, we can identify a supervisory signal that surely influences learning in the brain (Shapiro, 2012; Gopnik et al., 2020).

While these supervision signals largely differ from what is most commonly considered *supervised learning* in machine learning, we can still draw some parallels with human learning. We learn to categorise many concepts and objects as children, but most people carry on learning new categories as adults. For example, some people put effort in improving their understanding of the natural world by learning to recognise and name trees, plants or birds. Those who have engaged in such an endeavour may have noticed that the learning process is easier and faster if we count upon the expert knowledge of a friend or of technology such as *iNaturalist* (Van Horn et al., 2018). Another example: those

who have—or attempted to learn—a new language as an adult may have realised that whereas it is possible to learn the meaning of a new word by repeated exposure to it in multiple contexts, it is certainly easier if we look up the ground truth definition in a dictionary or, even easier, if there exists a direct mapping to a word in our native language. Summing up, not only does supervision facilitate learning, but human beings actively seek for it.

Besides this kind of explicit supervision, the brain certainly makes use of more subtle, implicit supervised signals, such as temporal stability (Becker, 1999; Wyss et al., 2003): The light that enters the retina, and the sound waves that reach the cochlea, are not random signals from a sequence of rapidly changing arbitrary photos or noise, but highly coherent and regular flows of slowly changing stimuli, especially at the higher, semantic level (Kording et al., 2004). At the very least, this is how we perceive it and if such a smooth perception turns out to be a consequence rather than a cause, then it should be a by-product of a long process of evolution that would be worth taking into account.

## 2.3 The role of evolution and brain development

In the previous sections, we have discussed some misconceptions or overstatements about how humans learn and generalise that are often found in the machine literature. Namely, that humans are able to generalise from a few examples and that this occurs with little or no supervision. Still, the commonplace comparison of artificial neural networks with human learning and the brain often misses a fundamental component of biology, recently brought to the fore by Zador (2019) and Hasson et al. (2020), although considered since the early days of artificial intelligence (Turing, 1968): the role that millions of years of evolution have played in developing the nervous systems of organisms in nature, including the human brain.

The most common way of training artificial neural networks, especially in machine learning research, is from *tabula rasa*, that is from randomly initialised parameters[1]. In contrast, a large part of the brain connectivity is encoded genetically and certain properties and behaviour are known to be innate, that is developed without prior exposure to stimuli (Farroni et al., 2005; Spriet et al., 2021). Importantly, evolution not only provides innate behaviour, but also determines what cannot be learnt, or relevant constraints—scientists who have trained animals in the laboratory for psychological or neuroscientific studies are well aware that tasks have to be carefully adapted to the ecological behaviour and limitations of the animal, determined by evolution.

Taking into account the role of evolution, we can draw conclusions that relate to the claims discussed in the previous sections. If our brains are the product of millions of years of exposure to relevant stimuli and adaptation, is it really fair to say that humans are capable of robust out-of-distribution generalisation and that we learn from from a few examples? If evolution has largely determined what our brain can and cannot learn, providing as with a "pre-trained model", is it really fair to say that humans learn in a unsupervised fashion? This questions are relevant for machine learning research: if we take biological learning as motivation for artificial intelligence, should we not temper our expectations of what learning algorithms should aspire to? And, therefore, would it not be worth reconsidering some research programmes?

On the flip side, insights from evolutionary theory are likely to be a fruitful source of inspiration for machine learning (Hasson et al., 2020; Zador, 2019). As we have observed, training a neural network from scratch may be more similar to a simulation of evolution than to the process by which an adult learns a new concept. As a shortcut to simulating evolution, neuroscience is a rich source of inspiration of constraints and inductive biases that determine learning in the biological brain and can potentially inform machine learning (Hassabis et al., 2017; Lindsey et al., 2019). For instance, simulating properties of the primary visual cortex in the early layers of an artificial neural network has been shown to improve adversarial robustness (Dapello et al., 2020; Malhotra et al., 2020).

Besides evolution, the focus on the capabilities of adults often makes us miss another important aspect of biological learning, particularly important in humans: the role of learning in infancy and brain development. While learning occurs too in adulthood, childhood is a particularly important and

---

[1]Some interesting and promising areas in machine learning research deviate from this standard approach. For example, transfer learning and domain adaptation study the potential of features learnt on one task to be reused in different, related tasks (Zhuang et al., 2019), and continual learning studies the ways in which machine learning models can indefinitely sustain the acquisition of new knowledge without detriment of the previously learnt tasks (Mundt et al., 2020). These approaches are inspired by biological learning or share interesting properties with it.

active time for learning (Atkinson, 2002; Gelman & Meyer, 2011). In fact, sensitive or critical periods for learning in infancy have been described or hypothesised, for example for vision (Harwerth et al., 1986) and language development (Lenneberg, 1967). Machine learning papers that draw motivation from the alleged generalisation capabilities of humans often underestimate the amount of input stimuli and supervision that infants receive (Gopnik, 2020). However, childhood can be regarded as period dedicated almost exclusively to learn, not only formally from parents and teachers, but also through playing, which *plays* a critical role in cognitive development Burghardt (2005); Pelz & Kidd (2020). Finally, the fact that humans—and other cognitively advanced animals, such as corvid birds, which also exhibit cultural learning—have a comparatively long childhood period, has led Uomini et al. (2020) to recently proposed that extended parenting is pivotal in the evolution of cognition. This adds to the discussion on the undervalued role of supervision. In sum, we propose machine learning research can benefit from drawing inspiration from both evolutionary biology and the literature on developmental psychology, brain development and life history and learning (Gopnik et al., 2020).

## 3 Supervision in machine learning

If we open a machine learning textbook (Murphy, 2012; Abu-Mostafa et al., 2012; Goodfellow et al., 2016), we will most surely find a taxonomy of learning algorithms with a clear distinction between *supervised* and *unsupervised* learning. However, while this separation can be useful, the boundaries are certainly not clear. As a matter of fact, if we take a look at the deep learning literature of the past years, we will also find abundant work on some variants supposedly *in between*—semi-supervised learning, self-supervised learning, etc.—whose definitions are all but clear.

### 3.1 Catastrophic forgetting of old concepts

If we recall a classical result in statistical learning theory and inference, the *no free lunch* theorem (Wolpert, 1996), no learning algorithm is better than any other at classifying unobserved data points, when averaged over all possible data distributions. Therefore, we need to constrain the distributions or, in other words, introduce prior knowledge—that is *supervision*. Recently, Locatello et al. (2018) obtained a related result for the case of unsupervised learning of disentangled representations: without inductive biases for both the models and the data sets, unsupervised disentanglement learning is fundamentally impossible. These results are purely theoretical and have limited impact on real world applications (Giraud-Carrier & Provost, 2005), precisely because in practice we use multiple inductive biases and implicit supervision, even when we do so-called unsupervised learning.

In a strict sense, even the classical, *purely* unsupervised methods, such as independent component analysis or nearest neighbours classifiers, make use of inductive biases, such as independence or minimum distance, respectively. Without inductive bias, learning is not possible: purely unsupervised learning is an illusion. While this is not news, the terminology used in the recent and current machine learning literature seems to reject supervision and neglect these nuances, evidencing that the field suffers catastrophic forgetting of well-established notions.

### 3.2 The brands of *alt-supervised* learning

Particularly in deep learning and computer vision, the term *supervised* learning has adopted, in practice, the meaning of *classification* of examples annotated by humans, that is models trained on examples labelled according to, for instance, the object classes. This is yet another instance of catastrophic forgetting—or, at best, abuse—of well-established concepts. It should not be necessary to recall that, first of all, *supervised learning* is a broader category than *classification*, which includes also regression and ranking, among other learning modalities. Second, even if we narrow our view to classification only, supervised learning is not restricted to learning from examples annotated by humans. Goodfellow et al. (2016) did not overlook this in their definition of supervised learning: "In many cases the outputs $\mathbf{y}$ may be difficult to collect automatically and must be provided by a human 'supervisor,' but the term still applies even when the training set targets were collected automatically".

In turn, the term *unsupervised* learning is now used for any model that does not use manually collected labels, regardless of what other kind of supervision it may use. Further, the term *semi-supervised* learning generally refers in practice to models that are trained with a fraction of the labels, but are tested on the same classification benchmarks. Finally, the term *self-supervised* learning has recently

gained much popularity, referring to models that are trained on tasks other than the standard task defined by classification labels.

Some of the methods proposed under these categories are certainly useful—that is not the subject of criticism of this work—but the terminology is overly confusing and unnecessary. A newcomer would easily fall into a scientific rabbit hole trying to discern the meaning of each of these names through publications—not to mention if they incorporated social media discussions into their endeavour. By way of illustration, the authors of this paper have witnessed how a recurrent question by students who learn about recent deep learning methods is whether there is any difference between *self-supervised* and *unsupervised* learning. Are students missing something fundamental? The following anecdotal recall of influential keynote talks at artificial intelligence conferences should shed some light on part of the origins of this confusion: In December 2016, Prof. Yann LeCun titled his NeurIPS keynote presentation "Predictive Learning", to refer to "what many people mean by unsupervised learning" (LeCun, 2016). A few years later, in his keynote presentation at ISSCC in February 2019, he spoke about similar ideas, but this time the title was "Self-Supervised Learning" (LeCun, 2019). In social media, he wrote: "I now call it 'self-supervised learning', because 'unsupervised' is both a loaded and confusing term" . Students may be getting things rather right.

Is there then a fundamental difference—a theoretically grounded one—between the deep learning methods labelled as *unsupervised* learning and more recently *self-supervised* learning? We argue that these are mostly brand names that reflect trends in the field, adding noise to the scientific progress and leading many astray. Therefore, we propose that, given the recent progress, the field of machine learning research would benefit from an exercise of self-reflection and from an effort to devise a rigorous taxonomy of the variety of methods. From a theoretical point of view, both the conventional classification models and the recent wave of self-supervised tasks can all be formalised as sub-categories of supervised learning.

## 3.3 Supervision comes in different flavours

In Section 2.2, we have seen examples of different forms of supervision used by humans and other animals. In machine learning, the field focused for many years on a few loss functions, such as classification and simple forms of regression. The relatively recent explosion of deep learning has brought about the development of several libraries for automatic differentiation (Baydin et al., 2017), which in turn have enabled the proposal of multiple loss functions and learning tasks with various types of supervision that can easily be optimised numerically by stochastic gradient descent and artificial neural networks. This has certainly opened promising and already fruitful avenues to incorporate richer forms of supervision and inductive biases other than classification, some inspired by biological learning, into machine learning algorithms.

A currently popular example is image data augmentation: Although until recently it was seen as a naïve technique to simply create additional training data, data augmentation actually encodes rich prior knowledge about human visual perception, in the case of computer vision. This is why it outperforms explicit regularisation methods, which provide less effective inductive biases Hernández-García & König (2018), and was used in "semi-supervised" tasks Laine & Aila (2016). The rich information embedded in image transformations has been used to encourage invariant outputs under different augmentations through contrastive losses (Ye et al., 2019), and even at intermediate representations, inspired by the invariance in the visual cortex (Hernández-García et al., 2019), although these methods were not branded as *self-supervision*. The use of this term for losses based on data augmentation was further popularised after the success of similar methods such as SimCLR (Chen et al., 2020). Beyond data augmentation invariance, the zoo of self-supervised learning tasks in computer vision is rich and diverse: classifying the rotation applied to image patches (Gidaris et al., 2018), predicting image colourisation (Larsson et al., 2017), classifying the relative position of two image patches (Doersch et al., 2015), or even solving full jigsaw puzzles (Noroozi & Favaro, 2016) (Jing & Tian (2020) recently performed an extensive review).

The current trend is to refer to these methods as *self-supervised* learning, but similar methods were referred to in the past as *semi-supervised*, *unsupervised*, and even *predictive* learning, as we have seen. A look at the papers reveals that these terms have been used mostly interchangeably. The terms *self-* and *semi-* and *un*supervised learning imply that *less* supervision is used, but it would be misleading to seriously argue that the tasks are devoid of supervision. Most of these techniques make use of a wide range of surrogate tasks with supervisory signals defined by humans. In fact, they could have been

called *hyper-supervised*[2] learning. Here, we contend that these methods are all variants of supervised learning, only that supervision comes in different flavours, both in biological and machine learning, and we should call it by its name and ideally develop a rigorous taxonomy.

# 4 Discussion

In this paper, we have discussed some of the overambitious promises of the deep learning hype, namely that machine learning models should be able to generalise to unseen distributions, from a few examples, without human intervention or supervision. These claims have often been motivated by alleged generalisation capabilities of humans. In order to assess these motivations, we have first reviewed, in Section 2, some often overlooked characteristics of biological learning relevant to machine learning research. In particular, we have argued that humans and other animals receive extensive and diverse input stimuli as well as multiple supervisory signals, including the long history of evolution and cultural transmission. In the light of these insights from biological learning, we have then, in Section 3, critically reviewed the various terms that are currently used to refer to supposed alternatives to supervised learning: semi-, self- and unsupervised learning, among others. In sum, we pointed out that all these approaches are in fact supervised learning—though not necessarily classification—and the machine learning (research) community would benefit from using more rigorous, less overselling nomenclature, and from devising a more rigorous taxonomy.

Supervision is not evil. It is at the core of statistical learning theory: learning is impossible without inductive biases or supervision. But supervision comes in different flavours, not only as classification labels. Neither is deep learning some sort of exceptional solution to learn without human intervention and supervision, nor is it a hopeless model class because it requires large data sets (Marcus, 2018). The human visual system is exposed to a lot of stimuli too. One exceptional advantage of deep learning is precisely that it is possible to effectively optimise different learning objectives, almost end-to-end, from large collections of nearly naturalistic sensory signals, such as digital images (Saxe et al., 2020). While other models are known to scale poorly as the amount of data increases, neural networks excel at fitting the training data and interpolating on unseen examples (Belkin et al., 2019; Hasson et al., 2020). This is a feature, not a bug. But we will make better progress if we exploit these advantages of deep learning without neglecting that supervision will always be necessary—the critical goal is how to best incorporate it and exploit it.

In this regard, we argue that deep learning needs *more supervision*, and not less. A major focus of the deep learning community in the last decade has been image object classification. This has brought about unprecedented progress and unveiled the limitations of having classification as chief task and class labels as main supervisory signal. For example, deep classifiers have been found to learn spurious features that are highly discriminative for the classification task but with little true generalisation power and clearly not aligned with perceptual features (Jo & Bengio, 2017; Wang et al., 2019; Geirhos et al., 2020). In fact, this mismatch has been argued to be at the root of adversarial vulnerability (Ilyas et al., 2019) and seems to be the consequence of training highly expressive, over-parameterised models in heavily unconstrained tasks. This can be addressed with meaningful constraints, that is more and richer supervision, possibly inspired by human perception and biological learning. For example, combining a classification loss with a similarity loss inspired by the invariance in the visual cortex yields more robust representations without detriment to categorisation (Hernández-García et al., 2019), and simulating the properties of the primary visual cortex may improve the adversarial robustness of neural networks (Dapello et al., 2020). Expanding in this direction leads to biologically-inspired, multi-task and representation learning, and away from just classification.

# 5 Conclusions for future research directions

The chief goal of this paper is rather descriptive than prescriptive. We have aimed to identify and describe aspects of the current trends in machine learning research that could be improved, in the hope of inspiring future work that effectively address them. Nonetheless, throughout the paper we have made suggestions that may help mitigate the confusion with the terminology, clarify research directions and ultimately bring about scientific progress in machine learning research. We outline these suggestions here to conclude the paper.

---

[2]The authors explicitly discourage the addition of a new term to the already too confusing list.

We have drawn parallels from cognitive neuroscience to contend that learning in nature also requires abundant data and supervision in multiple forms. Even evolution can be regarded as an optimisation process where natural selection is the supervisory signal. We have argued, as others have before, that these insights from biology, neuroscience and developmental psychology, among other fields, offer a great opportunity for machine learning research to draw inspiration and calibrate its compass.

As we have discussed, research in deep learning has departed from pure classification and has been exploring new learning tasks and ways of training artificial neural networks. Nonetheless, in some fields such as computer vision, the ultimate benchmark to assess the value of a method is still the accuracy on classification data sets, such as ImageNet, even though there is evidence of overfitting the test set. While object recognition will remain an important benchmark, as deep learning is well suited to learn representations, we should develop methods to assess the quality of the learnt representations for tasks other than classification. In this regard, we encourage researchers to evaluate their models with tests that are still not widespread, such as the suitability for transfer learning, adversarial robustness, comparison with brain measurements, behavioural tasks, etc.

We have also argued that the field would benefit from an effort to devise a rigorous taxonomy of learning methods that sheds light on the ocean of methods proposed in the past years. The terms *self-*, *semi-* and *un*supervised learning have been used interchangeably and this is often a source of confusion for students and newcomers. While confusing terminology is natural in a rapidly growing, the time might have come for distilling the progress of the past years into rigorous nomenclature that better survive the test of time.

Finally, we recall that most of the learning theory has been developed for simple loss functions such as binary classification or mean squared error regression, but certain methods successfully used in practice today escape the available theory. Given the success of this kind of more complex supervised objectives, the study of these methods from a theoretical point of view might be a fruitful direction for future work.

## Broader Impact

Since this article does not present a new method or results from data sets, potential risks of "bias in the data" or "failure of the system" do not apply. As a critical review of current trends in the field and cite multiple research articles, some researchers could potentially feel addressed and affected by our mentions. We declare that we do not intend to negatively affect any individual researcher and we have only referred to individuals directly in the case of well-established scientist with a reputation. Our goal has been in any case to potentially improve scientific progress through a constructive reflection.

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
