# OpenReview forum: "Rethinking supervised learning: insights from biological learning and from calling it by its name"
_NeurIPS.cc/2021/Conference — NeurIPS 2021 Submitted_

### Official Review · Reviewer_6UDa · 2021-07-13

**Rating:** 6
**Confidence:** 4

**Summary:**

With this opinion paper the authors aim at clarifying some terminology commonly used in the modern deep learning literature. The goal is to highlight the strong overlap between several approaches that have been proposed as "alternatives" to supervised learning: predictive, unsupervised, semi-supervised and self-supervised learning. According to the authors, such proliferation of terms has been promoted by the (wrong) conviction that supervised learning is too limited in scope and does not account for the great variety of learning mechanisms observed in biological agents. The authors argue that the deep learning community would benefit from a clarification in terminology, which could highlight that, overall, all learning approaches can be defined as “supervised”.

**Ethical Concerns:**

I do not see ethical issues related to the content of this paper.

**Limitations And Societal Impact:**

The authors did not explicitly discuss the limitations of their research approach, but being this an opinion article I think this issue is less relevant.

**Main Review:**

Originality: The perspective advocated by the authors is provocative and original. It addresses an important topic with broad vision and well-motivated arguments.

Quality: I enjoyed reading this paper. Though (to the best of my knowledge) opinion papers are not common in NeurIPS, I believe the subject of the article is relevant and timely, since it is often a matter of heated discussions with colleagues at AI conferences. I think it is very useful to clarify common terminology and try to organize the taxonomy of modern machine learning approaches. Maybe this would not be tremendously useful for experts and people working in the field from a long time, but it could be an important guide for newcomers and students. At the same time, however, I feel that the paper is sometimes a bit superficial and could be improved.

The most critical point, in my opinion, is that the discussion should be accompanied with more technical definitions: the different approaches mentioned by the authors as “alternatives” to supervised learning do have peculiar features that justify calling them with specific names, and such features should be clearly explained if the aim is to clarify the terminology. For example, semi-supervised learning entails that some labels are directly provided by human experts, while self-supervised learning usually refers to the setting where the agent discovers by itself the relevant categories, for example by exploiting related tasks. Unsupervised and predictive learning, instead, usually do not involve any label, and the “supervision” process is mostly related to the design of proper distance metrics or “introduction of prior knowledge” (Line 216).

I think that most of the confusion stems from the ambiguous definition of “supervision”, which is normally associated with the use of explicit labels (discrete classes or real numbers) to characterize each input pattern. In this paper, instead, the authors seem to associate the term “supervision” with the more general notion of “feedback”. According to their view, “supervision” just implies that the agent can obtain some useful learning signals, for example through active interaction with the environment. If interpreted in this sense, I agree with the authors that all learning approaches should be considered supervised. However, I think that the most common way to think about “supervision” is to assume that the emerging semantics in the model should align with the conceptual semantics externally imposed by humans (i.e., classification, regression and ranking). If taken in this sense, I do not agree that all learning approaches should be considered supervised.

It would also be great if the authors could more explicitly put each learning approach in relation with biological learning principles, rather than just generically saying that biological learning and evolution imply lot of supervision signals. For a relevant discussion about the role of supervision and top-down feedback in cognitive models, the authors could refer to Testolin and Zorzi (Frontiers in Computational Neuroscience, 2016), which use “generative learning” as yet another alternative term that could be disambiguated. In relation to predictive learning, the authors could refer to Friston (Nature Reviews Neuroscience, 2010) and Clark (Behavioral and Brain Sciences, 2013), which provide a wealth of biologically-rooted perspectives on this powerful learning modality.

Few typos:
-	Line 177: these
-	Line 201: propose
-	Line 366: growing…

Clarity: The paper is written in a very clear way and properly organized.

Significance: Deep learning researchers can benefit from a thoughtful clarification of relevant terminology and for a condensed overview of how state-of-the-art models incorporate biological learning principles.


**Time Spent Reviewing:**

6

---

> ### Author Response · Authors · 2021-08-10
> **General response to the review**
>
> Dear Reviewer 6UDa,
>
> We thank you first of all for reviewing our paper and providing insightful feedback. We particularly appreciate that you have provided an assessment of each dimension from NeurIPS's guidelines, namely originality, quality, clarity and significance.
>
> We are also pleased by your positive feedback and assessment of our paper: that you think it tackles and "important topic with broad vision and well-motivated arguments", that it "could be an important guide for newcomers and students", that you "enjoyed reading" and think it is "written in a very clear way and properly organized". We also thank you for identifying some typos and pointing to relevant papers.
>
> We will tackle next the weaker points that you identified and the concerns that you raised.
>
> * "more technical definitions [of the _alternatives_ to supervised learning]": we agree that the terms reflect, to a certain extent, some nuanced differences in different approaches. The reason why we do not provide technical definitions of each term is precisely because we argue it would be impossible to be accurate and technical. All we can do is give definitions that reflect the use of the terms in the literature according to our review. We attempt to do this in Section 3.2. We take your feedback and will improve this section to be more precise in what we mean by each term and illustrate the points with more references from the literature. However, the impossibility of providing precise definitions of each of these terms is one of the points we make in our paper (see the paragraph starting in line 259). In our literature review, we have come across several papers that have attempted to provide definitions of self-supervised learning, semi-supervised learning and unsupervised learning. Our conclusion is that 1) the definitions are not consistent across papers and 2) in some cases the definitions are not consistent even within the same paper. We decided not to discuss these papers (which would have helped us make the point), in order to prevent any negative consequence on the authors of those papers. Another telling observation from our literature review is that many papers actually use the three terms interchangeably (within the same paper) for the same technique proposed in the paper, as though it was important to include these terms to position the contribution of the paper. We have also decided not to explicitly point at these papers as examples of questionable practices.
> * "ambiguous definition of _supervision_": from our review of biological learning in Section 2, we argued that living organisms use multiple "supervisory signals". In most machine learning papers, "supervision" is "normally associated with the use of explicit labels", as you wrote. We titled our article "Rethinking supervised learning", implicitly suggesting that our understanding of machine learning would benefit from a broadening of the meaning of "supervision", closer to the idea of "supervisory signals" in biological learning, which can be rich and varied. In this regard, we do use the term "supervision" as related to the more general notion of "feedback", directly answering your question. What we also argue in the paper is that having sub-categories of the types of algorithms and methods (a taxonomy) is useful to map and organise our understanding, but it should be fundamentally grounded, as opposed to the vague and liquid definitions of self-supervised learning, for instance. We encourage future work, ideally collaborative, to propose such a taxonomy. Stemming from this piece of feedback, we will make an effort to clarify the use of the word supervision in the paper.
> * We agree with the idea of more explicitly connecting learning modalities with biological learning, as well as including a discussion of the specific role of supervision in learning in the brain. Thank you for pointing to these relevant papers, as we were not familiar with all of them. We will update the paper (Section 2 and 3 specifically) to include this discussion, since we left some space for incorporating feedback from the reviewers.
>
> We hope we addressed your concerns and comments and look forward to discussing further during the discussion period. Thank you again for the insightful review.

---

> ### Author Response · Authors · 2021-08-27
> **Gentle ping**
>
> Dear Reviewer 6UDa,
>
> We understand that most reviewers are heavily loaded and the timing of the discussion period is not ideal, being a popular time for vacation in many places around the world.
>
> We would just like to thank you again for your review and kindly ask you whether you had the chance to read our rebuttal. In that case, we are eager to know whether you concerns were addressed or others remain.

---

> > ### Comment · Reviewer_6UDa · 2021-08-28
> > **Feedback following rebuttal**
> >
> > Dear Authors,
> >
> > I carefully read your responses and appreciated your effort in improving the paper and engaging in a discussion with the Reviewers. At the same time, in line with the general opinion of the other two Reviewers, I am still not convinced that this would be a strong "NeurIPS" contribution. Maybe a more cognitively-oriented venue could be more appropriate for discussing these ideas.

---

> > > ### Author Response · Authors · 2021-08-29
> > > **Short question**
> > >
> > > Thank you for the feedback. Besides the general statement, do you think your specific concerns were addressed in our rebuttal? Namely, regarding the definitions of modalities of learning, the clarification of the use of "supervised learning", and the more explicit connection with biological learning principles and the specific references suggested in the review.
> > >
> > > PS: Regarding the scope and the appropriate venue, our paper is heavily focused on machine learning topics. We review literature from cognitive science and neuroscience which is relevant for machine learning, not the other way around. Therefore, we don't think our paper would be suitable for a cognitively-oriented venue, but rather for the machine learning community.

---

> > > > ### Comment · Reviewer_6UDa · 2021-08-30
> > > > **More detailed feedback**
> > > >
> > > > I think that my concerns were only partially addressed in your rebuttal. I appreciate your willingness to clarify the use of "supervised learning" and to more explicitly mention the relevant biological/cognitive literature. However, I still feel the contribution is a bit superficial: I know it can be hard to exhaustively include technical definitions along with a more general discussion, but I really think that terminology should be clarified with precise definitions in order to make the paper useful for a broad audience. I believe the authors should strive to be as accurate and technical as possible in their writing: if you found inconsistent definitions in the literature, a strength of your paper could be to try to clarify them as much as possible.
> > > >
> > > > Moreover, in partial alignment with the opinion of Reviewer pJTi, I also feel that your work seems mostly aimed at educating the audience and does not offer deep insights, despite it raises original and provocative points. In this respect, your review could be mostly valuable for newcomers and students, and this is why I do not think that NeurIPS is the proper venue to publish this article (though I agree with Reviewer pJTi that the "NewInML" workshop could be an interesting place to discuss these ideas and receive feedback from colleagues).

---

> > > > > ### Author Response · Authors · 2021-08-30
> > > > > **Thanks**
> > > > >
> > > > > Thank you for elaborating on your assessment.

---

### Official Review · Reviewer_YQEE · 2021-07-16

**Rating:** 2
**Confidence:** 5

**Summary:**

The manuscript offers perspectives on machine learning and connects some general claims in ML with some findings in cognitive science. The authors make four main suggestions for moving forward in the field (see below for a summary of contributions).

**Limitations And Societal Impact:**

Yes

**Main Review:**

The authors examine important claims in machine learning. However, in its current state, this manuscript reads more like a high-level blogpost than a paper that would appear as a full publication in a top ML conference. The manuscript can be substantially improved if the authors provide more evidence for their claims about ML research directions. It would be especially beneficial to back up their argument with some statistics that can quantify the trends they describe instead of solely relying on quotes from a handful of papers ( for example, some statistics about number of papers on a certain topic with a certain claim). Below are some more specific issues that I have with the manuscript:

1. The exposition is filled with unnecessarily inflammatory language. Some examples just from the first 50 lines: L4 "gave rise to a hype loaded with ambitious promises and overstatements", L6 'supervised learning went, for many, from glory to shame", L49 "based on misconceptions and overstatements about biological
47 learning, and amplified by overselling nomenclature". The manuscript would benefit from toning this down and allowing factual evidence to make its point. Here is the one that gave me the most pause: L180 "would it not be worth reconsidering some research programmes?". This seems to be completely the opposite of
the authors' self-described goal of being descriptive rather than prescriptive.

2. The authors ​dismiss important challenges to modern ML by unfairly equating them to much more extreme versions of these challenges in human intelligence. For example, the authors dismiss the importance of addressing out-of-distribution generalization in ML by saying that humans are also limited in their
ability to transfer to new domains, as exemplified by the difficulty to learn a new unrelated language. Out-of-distribution generalization comes in many challenging forms to ML systems (e.g. changes in the label distribution (P(Y)), changes in the input feature distribution (P(X)), and changes in P(Y|X))). The example of humans learning a different language is an example of changing all of these distributions, which is an extreme form of out-of-distribution generalization. ML systems are shown to be sensitive to much milder changes in the problem setting.

3. The contributions of this manuscript are not clear. It's not clear to me what this manuscript offers as a conclusion that is not already pursued in ML.
- Two of the four conclusions that are drawn -- that evaluation beyond singe task performance is important and that a theoretical understanding of complex supervised objectives is important -- have been steadily growing directions in the field for over 5 years now.
- One of the other 2 conclusions is that ML should temper its expectation about how sample efficient learning should be because humans are exposed to more supervision than is suggested in the ML literature. The authors oversimplify the aims in many of these ML works.
ML researchers don't expect that an ML system will develop abilities to generalize to a new class out of thin air. Of course a system needs to be exposed to "relevant samples" in order to learn. Not all of these samples however have to come from the specific class.
The hope is that a sample efficient ML system will be able to generalize from previously learned related examples in order to support fast learning of new examples.
- The last remaining conclusion is that it would be great to distill progress in ML methods into a standardized nomenclature. I agree, as do many other ML researchers. A fitting contribution for a venue such as NeurIPS would be to actually suggest such a nomenclature. I don't consider a call for such nomenclature to be enough of a contribution.




**Time Spent Reviewing:**

3

---

> ### Author Response · Authors · 2021-08-10
> **Response to the general review**
>
> Dear reviewer YQEE,
>
> Thank you for your time in reviewing our manuscript. We are, of course, not pleased (and humbly, also surprised) by having received the second worst overall score, but we will here address your comments, questions and concerns, both for the sake of acknowledgement and reciprocity and because we are confident that we may be able to convince you that our manuscript may deserve a higher consideration.
>
> In your review, there are three specific issues that we will address below specifically and separately. Otherwise, we deduce from the general paragraph of review that the overall score of strong rejection should have to do with the lack of statistics to quantify the trends in machine learning research that we describe in your paper, as it is the only argument in the paragraph besides the comment about our paper reading "more like a high-level blogpost than a paper". The latter is a rather subjective statement and we are unsure about how to receive it or address it. Could you please elaborate on this point? We will therefore focus on the former argument, namely the lack of statistics.
>
> We agree that it would be interesting to see statistics about the trends in machine learning research and we indeed take a note of this piece of feedback for our future work. However, we disagree in this being a major weak point in our paper. In fact, offering a quantitative analysis of the trends would shift the focus on that analysis and its conclusions and it would shift the attention from the points we are making. It would be substantially different paper. Ours is a perspective or opinion paper, written for the audience formed by the machine learning research community that attends NeurIPS (where our paper is submitted). We assume that this audience is reasonably familiar with the current trends in their community, which we believe is a fair assumption. Nonetheless, we do not simply make that assumption and go straight to our core analysis, but we rather offer a historical perspective and review of influential work that has shaped the field, in order to build and support our arguments. Please note that we deliberately decided not to critically discuss specific papers except those by prominent and well-established members of the field, as the opposite could have negative consequences for the authors and is not our intention. We believe that we provide an in-depth review of the literature (both in machine learning and other sciences) to support our motivation and analyses. Do you think that our paper is not well motivated in the introduction, and that the paper could only be justified by including a quantitative analysis of the trends, assuming this is technically possible?
>
> We next address the three remaining specific issues raised in the review:
>
> 1. "The exposition is filled with unnecessarily inflammatory language": This seems, as well, a subjective observation. If the language is indeed "inflammatory" and if so, "unnecessarily" so, is hard to assess objectively. We could say that the language in the abstract and the beginning of the introduction (especially, but not only) may be (deliberately) _uncommon_ in NeurIPS submissions, as this is an opinion or perspective paper, not common at this venue. Is that necessarily a condition for rejection? Which of the criteria for assessing NeurIPS submissions (originality, quality, clarity and significance) is affected by our choice of vocabulary? We argue that, as an opinion or perspective paper, adapting the language helps in the transmission of the message, and therefore improves its clarity. As a matter of fact, if our choice of language has triggered a reaction in the reviewers (in both directions), we could say that we have succeeded in catching the attention of the reader. We intended to make our paper easy to read and even entertaining, which according to other reviewers has been the case. Our choice of language also has to do with this intention.
>
> We will briefly comment each of the sentences that you highlighted in the review:
>
> * L4 "gave rise to a hype loaded with ambitious promises and overstatements": we discuss these as early as in the introductions. We were however not very innovative in using the word _hype_ for deep learning though.
> * L6 'supervised learning went, for many, from glory to shame": this anticipates specifically our discussion in Section 3.2, where we discuss the trends and terminology that has emerged as a rejection of _supervised learning_ (technically, it should be _classification_, as we discuss).
> * L49 "based on misconceptions and overstatements about biological 47 learning, and amplified by overselling nomenclature": the first part refers to Section 2, where we amply elaborate on the misconceptions and overstatements about biological learning. The second part refers to Section 3, where we analyse the terminology used.
>
> "The manuscript would benefit from toning this down and allowing factual evidence to make its point": We appreciate your feedback, but this is subjective and we humbly disagree. As discussed above, we believe that using uncommon language in the introduction helps capture the reader's attention and we indeed provide factual evidence for our statements, in the form of a literature review (including but not limited to quotes).
>
> 2. "The authors dismiss important challenges to modern ML by unfairly equating them to much more extreme versions of these challenges in human intelligence. For example, the authors dismiss the importance of addressing out-of-distribution generalization in ML by saying that humans are also limited in their ability to transfer to new domains": Where exactly do we _dismiss_ the importance of addressing out-of-distribution generalisation in ML? We make the point that often in machine learning papers, humans are put as examples of systems capable of out-of-distribution generalisation. We then amply discuss how these claims are largely overestimated, again by providing an in-depth review of works in neuroscience and cognitive science, that may be unknown to part of the machine learning community. We cannot dismiss the importance of such part of ML research, as we actually work on it. We, however, claim that we could benefit from better understanding learning and generalisation in biology. Do you disagree with this? We believe that there might have been a misunderstanding of this aspect. We would appreciate if you could point to the specific parts where we may _dismiss_ OOD research. Regarding the specific example of language, we could argue that the difference between two languages is relatively small in terms of the data distribution. In fact, ML systems have been shown to largely benefit from transfer learning in multi-language tasks, arguably more than humans. Of course, we agree with the reviewer that ML systems are often affected by mild changes, and that is an important area of research. We do not think that our paper dismisses it, as discussed, but the opposite.
>
> 3. "The contributions of this manuscript are not clear. It's not clear to me what this manuscript offers as a conclusion that is not already pursued in ML": It seems that you have mistaken the section (5) "Conclusions for future research directions" as our contributions. This is simply one section of the paper, one that aims at distilling the rest of the paper into some general ideas for future research directions. But our contribution is not limited to these conclusions. The main contribution of the paper, especially as an opinion or perspective paper, is the body of the paper itself. For example, the central section of the paper is Section 2, "Supervision in biological learning", where we provide "insights from biological learning", as summarised in the title. Here, we provide a review of learning and generalisation in biological learning, supported by multiple scientific references and examples that are likely to be unknown for many in the machine learning field. Is not this a valuable contribution of our paper? This review, moreover, put in context and related to current trends in machine learning research. We humbly argue that the discussion and review of the need for more relevant _training_ data in biological learning than is typically claimed (Section 3.1), the various supervisory signals used by animals (Section 2.2), and the role of evolution and brain development (Section 2.3), are novel and valuable contributions. Section 3 builds upon Section 2 by connecting the conclusions from biological learning to fundamental concepts in machine learning such as generalisation, supervision and inductive biases, and we offer a descriptive, (self-)critical discussion of current trends, with an emphasis on those that may be problematic, with the spirit of improving the research in our own field. We argue that this is also a valuable contribution. Therefore, our conclusions for future directions are only an additional piece of contribution, not the summary of our contributions. We hope that this has clarified your concern. Nonetheless, we would like to briefly address your comments about each specific conclusion:
>
> (_continues briefly in the next comment_)

---

> > ### Author Response · Authors · 2021-08-10
> > **Response to the general review (#2)**
> >
> > (_continuation from previous comment_)
> >
> > * "Two of the four conclusions that are drawn -- that evaluation beyond singe task performance is important and that a theoretical understanding of complex supervised objectives is important -- have been steadily growing directions in the field for over 5 years now.": As readers and reviewers (hence exposed to many reviews) of the major machine learning conferences, we can confidently say that the main and often only focus of machine learning papers is on single metrics on benchmarks. We actually believed that this is not an original critique and is shared by many in the field. The theoretical understanding of complex supervised objectives may have improved in recent years, but we are far from a satisfactory level of understanding. Do you agree? Therefore, we believe it is important to keep pushing towards these goals, which is the intention of this section.
> > * "ML should temper its expectation about how sample efficient learning": Your critique regarding this point is that we "oversimplify" the aims of some ML works. Could you please point out where in the paper (in Section 5?) we do this oversimplification. We are unable to locate the source of your critique. You enclosed "relevant samples" in double quotes as a pointer, but this is not written in the paper.
> > * "The last remaining conclusion is that it would be great to distill progress in ML methods into a standardized nomenclature": We are glad that you agree in this point, and we hope to encourage such future work. Do you agree with us that it would be a completely different paper if we included such proposal? We believe it is reasonable to leave this as future work. This is what we mean by being descriptive, not descriptive. Future prescriptive work will hopefully follow up descriptive work, ideally after discussion in the community. This is a healthy cycle in science, in our humble opinion. You write that you "don't consider a call for such nomenclature to be enough of a contribution". If that was all we did in our paper, we could not agree more, but we hope we don't need to explain that that was not it. Do you agree that this is an oversimplification of our contributions?
> >
> > By the way, in Section 5, we also outline that ML would benefit from a stronger connection with neuroscience, biology and developmental psychology, among other fields. This is the first conclusion and is connected to Section 2, as we have discussed, which is central. You seem to have excluded this from the "four conclusions".
> >
> > We hope that we have addressed all your concerns and comments from your review. We would be very grateful if you can follow up the discussion. Thank you again for the review.

---

> > > ### Comment · Reviewer_YQEE · 2021-08-18
> > > **clarification**
> > >
> > > I appreciate the authors' resolve and confidence in their manuscript. However, I still firmly believe that this paper needs to be drastically revised in order to significantly benefit the ML community. As the authors themselves clarify in the rebuttal, their contribution is the writing itself. Therefore the writing needs to be crystal clear. I agree that the evidence on the neuroscience and psychology side is specified and explained well, but the evidence on the issues with the ML side is not. The fact that two of the three reviewers have concluded this and urged the authors to include more tangible examples of these issues in ML besides quotes from a handful of paper expositions should prompt the authors to take this criticism seriously.
> > >
> > > One example of the oversimplification of the aims of some ML work is in Section 2.1: Generalisation requires exposure to relevant training data. Specifically, the lines: "In the argument that machine learning models should generalise from a few examples, there seems to be a promise or aspiration that future better methods will be able to perform robust visual object categorisation—for instance—among many object classes after being trained on one or a few examples per class." To which, my response was "ML researchers don't expect that an ML system will develop abilities to generalize to a new class out of thin air. Of course a system needs to be exposed to "relevant samples" in order to learn. Not all of these samples however have to come from the specific class. The hope is that a sample efficient ML system will be able to generalize from previously learned related examples in order to support fast learning of new examples."

---

> > > > ### Author Response · Authors · 2021-08-27
> > > > **Modification of the paragraph. Did we address the other concerns?**
> > > >
> > > > Than you very much for following the discussion of our manuscript. We highly appreciate your time.
> > > >
> > > > We definitely take your "criticism seriously", as well as that from the other two reviewers. This is why we have tried to carefully discuss every concern raised in your reviews, as well as we have translated some of your feedback into specific changes in the manuscript, which we have noted in the rebuttals. We are deeply grateful for such valuable feedback.
> > > >
> > > > Regarding the specific concerns that you raised, and that we individually replied in the rebuttal, it is not clear to us from the follow up message which aspects remain unresolved.
> > > >
> > > > We will here briefly discuss the "example of oversimplification of the aims of some ML" that point out in our paper:
> > > >
> > > > > [Paper] In the argument that machine learning models should generalise from a few examples, there seems to be a promise or aspiration that future better methods will be able to perform robust visual object categorisation—for instance—among many object classes after being trained on one or a few examples per class.
> > > >
> > > > > [Reviewer YQEE, in the original review] "ML researchers don't expect that an ML system will develop abilities to generalize to a new class out of thin air. Of course a system needs to be exposed to "relevant samples" in order to learn. Not all of these samples however have to come from the specific class. The hope is that a sample efficient ML system will be able to generalize from previously learned related examples in order to support fast learning of new examples."
> > > >
> > > > We obviously agree with your statement, and therefore we regret that this first paragraph in Section 2.1 may be interpreted as though we think otherwise. Thus, we will propose a modification of the wording aimed at clarifying our position (see below). Please, let us first note that this paragraph serves as an introduction to a section that is primarily aimed at highlighting the *often underestimated exposure to data in biological learning systems*. We illustrate this with a quote from a highly cited paper (Vinyals et al.): "a child can generalize the concept of 'giraffe' from a single picture in a book". Very similar statements are commonly seen in the literature, as we discuss in the paper. In our opinion, it is these statements that are indeed oversimplifications of the nature of biological learning and our purpose in this section is to shed light on this matter with a review of relevant literature from cognitive science, neuroscience, developmental psychology, evolutionary biology, etc.
> > > >
> > > > We propose the following paragraph as a modification to the current introductory paragraph of Section 2.1:
> > > >
> > > > > Machine learning articles about few-shot learning and out-of-distribution generalisation often motivate the need for better sample efficiency and generalisation capabilities by stating that humans learn and generalise from a handful of examples. While a primary objective is to develop techniques that efficiently extract the maximum possible information from the available examples, we should also remind ourselves that no machine learning algorithm can robustly learn anything that cannot be inferred from the data it has been trained on. Although this may seem to contradict certain current trends and statements in the literature, we should also bear in mind that learning in nature is not different, and humans and other animals are exposed to more data than it is often claimed.
> > > >
> > > > Note that we have modified the first and the last sentences of the paragraph. Since the paragraph now concludes with a similar sentence as the following paragraph, we now simply start the following paragraph with the second sentence:
> > > >
> > > > > A biological brain continuously receives, processes and integrates multimodal inputs from various sensors [...]
> > > >
> > > > We believe that this new version should make our intended point more clearly. Please let us know if you agree, and thanks again for the feedback.

---

### Official Review · Reviewer_pJTi · 2021-07-17

**Rating:** 3
**Confidence:** 5

**Summary:**

This article is a position statement that aims to point out weaknesses in the
authors' conception of the ML field's agenda. As the authors state, the work is
descriptive, not prescriptive: it does not provide specific directions, but rather
points to trends that (in the extreme) are problematic.
The key points in the article include: (1) natural environments provide rich
support for learning beyond traditional supervised classification labels; (2)
evolutionary and developmental processes provide strong inductive biases that can
inform model design; (3) the amount of data that natural learners have available
is often underestimated (as is one-shot learning abilities); (4) there are many
signals in natural environments which can constrain learning beyond traditional
class labels.

**Limitations And Societal Impact:**

yes

**Main Review:**

This work would be a provocative position paper for a workshop,
but I fail to see what research contribution it makes. The authors' caricature
of the field's naivete does not match my impression of the field (having been
an active researcher for over 30 years). They point to relevant psychological
and neuroscientific literature (which the authors do know well) that the
ML field would do well to be more aware of. Much of the literature cited has
been the subject of more cognitive science focused research at NeurIPS.
Like the authors, I wish the field would focus more on cognitive inductive bias
rather than building larger and larger systems with larger and larger data sets.
But the way to convince the field is not to lecture it, but to demonstrate that
the authors' insights into biology and cognition can be productively incorporated
into computational ML approaches. For example, in small data regimes, show that
cognitive biases (whether in architecture, objective functions, or data sources
available for learning) allow for what would otherwise be unlearnable.
The article also points to some boastful and unfortunate quotes from prominent figures,
but to shoot down the field for a few quotes is unnecessary. What the authors need
to make their argument convincing is specific examples of research done wrong and
research done right, rather than speaking in generalities. I do not believe
that the article in its current form would perturb any researcher from their
current trajectory.

The article reads as a stream-of-consciousness series of thoughts and contains many
dated and incorrect, or at least arguable, points, particularly with the authors' conception
of the 'supervised' and 'unsupervised' labels.  Let me illustrate with just one
particularly frustrating section, Section 3.1.

* The section is headed "catastrophic forgetting" yet the content of the section says
nothing about this classic issue in neural nets.  Did the authors change their mind
about what content to include in the section?

* The authors' take from the no-free-lunch theorem is incorrect: the fact that no
one model/architecture is superior to all others on expectation across data
distributions does not motivate supervised learning per se. It motivates choosing
inductive bias appropriate for a task. (The theorem assumes a certain objective function,
which could be either supervised or what is usually considered an unsupervised
criterion, such as cluster compactness.)

* Any well-formulated unsupervised learning method requires an objective function  and optimization. The authors appear to consider this a form of 'implicit supervision'.   Unsupervised learning does not mean that the algorithm does something random.
It means that the objective does not require guidance concerning specific responses from a teacher.   And unsupervised learning does not mean that the model is without inductive bias. As the authors state, "this is not news", but they then go on to claim (without support) that the field is unaware of these facts.

* Note that from a probabilistic perspective, unsupervised and supervised learning
share a common foundation. Rather than treating supervised and unsupervised
learning as two qualitatively different paradigms, every decent probabilistic ML textbook makes
the point that follows.  Unsupervised learning is about modeling a distribution
over observed features.  When those features are images and class labels,
learning a joint distribution over the observations will permit estimation of
conditional probabilities. Thus, from a probabilistic perspective, unsupervised
and supervised learning aren't qualitatively different methods; supervised
learning is a special case in which a model must estimate the _conditional_
distribution P(output|input), whereas
unsupervised learning does not split the data elements into 'input' and
'output' and thus requires modeling the _joint_ distribution, from which any
feature can be predicted from any other. One reason why the probabilistic
perspective is so popular in ML is because it provides this unified perspective.

* The authors state that supervised learning goes beyond classification to include
regression problems. Every textbook makes this point. Every ML course makes this
point.  The wikipedia 'supervised learning' page makes this point. I make this point in the first day when I teach ML. Again,
this is not news to anyone in the field. The authors may be expressing their frustration at
the predominance of research on classification, but there is no shortage of work on regression
problems at NeurIPS and other ML conferences.

* The authors seem to have a somewhat dated conception of the field. Sure, in 2012
image classification was the focus of the groundbreaking research.
One reason why image classification continues to be so popular is that the image
data sets are well understood, and these image data sets provide sensible test beds for
better understanding deep learning and exploring novel optimization methods, architectures, etc. However, the field is much
richer than the authors allow. For example, the introduction of the transformer
architecture for language processing--as horribly non-cognitive as it is--has become a
huge industry in and of itself. Video understanding and interpretation (not just
classification) can be found at every recent ML conference. Work on video-text interactions
is now moving to the foreground. The field is advancing and the problems researchers
choose to focus on changes over time.


**Time Spent Reviewing:**

3 hours

---

> ### Author Response · Authors · 2021-08-10
> **General response to the review**
>
> Dear Reviewer pJTi,
>
> We first thank you for reviewing your manuscript. We appreciate your time and your feedback.
>
> From your review, we understand that the main criticism of our paper is that you "fail to see what research contribution it makes" (initial sentence of the review), the style we used (main paragraph of the review) and that it "contains many dated and incorrect, or at least arguable, points" (remaining part of the review. We will address each of these concerns in our response.
>
> We are surprised to read that you failed to see the contributions of our paper because you pointed to some of them in the summary part of the review. As you noticed, this is an opinion, perspective or position paper. Therefore, the way it contributes to the field differs from we typically see at NeurIPS. It is not trivial to summarise the contributions of such a paper in a few words, but we would like to point you to the last paragraph of Section 1 Introduction, and the introductory paragraph of Section 4 Discussion, where we outline the main aspects of our paper. In short, we first provide a historical perspective of the trends in machine learning research in recent years to as a motivation (Section 1). Then, in Section 2, we offer an in-depth review of learning and generalisation in biological systems, citing relevant research from neuroscience and cognitive science that is likely to be both unknown and of interest for many in machine learning research. We contextualise this review for what is relevant for machine learning, as this has been submitted to a (mostly, today) machine learning venue. We more explicitly do so in Section 3, where we connect the conclusions from Section 2 with fundamental concepts in machine learning such as generalisation, supervision, inductive biases, etc. and with recent and current trends in the field, providing a (self-)critical review of those trends that we see as potentially problematic, with the aim of improving progress in our own field. We humbly believe that these are valuable contributions for the machine learning community.
>
> You seem to agree with our points regarding learning in nature. Don't you see our review of the literature in cognitive neuroscience about aspects that are relevant to machine learning as a contribution? In fact, the motivation of such review is our identification of multiple misconceptions about biological learning in machine learning papers. If you agree with our points, then don't you also think this review can be of value for many in machine learning, in order to gain better inter-disciplinary understanding? We realise that you, in particular, as an "active research in [machine learning] for over 30 years", and with an interest or sympathy for the cognitive neurosciences, may not have learnt a lot of form our position paper, and may even find unnecessary our critical review of recent trends, as you have solid foundations to not miss the forest for the trees. However, it would be trivial to argue that in a field that has grown exponentially in a few years, someone with 30 years of experience in the field is part of a remarkable minority. In fact, it would also be straightforward to argue that the majority of members in machine learning have a few years of experience only. The contribution of our paper is not new results or new data, but connecting insights from different fields, reviewing inter-disciplinary research, and identifying potentially problematic trends in our field. In your opinion, "the article in its current form would [not] perturb any researcher from their current trajectory". You may not need our paper, as you probably do not need many others published at NeurIPS, and it may not perturb your well-established trajectory, but have you considered whether other members of the community without 30 years of experience could gain insights from this paper? Your fellow reviewer 6UDa, in fact, puts it that way: they "think it is very useful to clarify common terminology and try to organize the taxonomy of modern machine learning approaches. Maybe this would not be tremendously useful for experts and people working in the field from a long time, but it could be an important guide for newcomers and students". Besides, they highlight the timely relevance of the topic: "the subject of the article is relevant and timely, since it is often a matter of heated discussions with colleagues at AI conferences"
>
> You also seem to have negatively received the "provocative" style. We also realise that it may not work with everyone, especially those with long experience in the field, but we argue that this is justified in such an opinion or perspective paper and it actually helps gaining the attention of the reader, as it has been acknowledged by other reviewers. We believe that there might be a misunderstanding in our intention with the paper, as you imply that we intend to "shoot down the field". Quite the opposite, we intend to point out several aspects that may be slowing down the field and leading many astray, and propose ideas to move forward, to hopefully help our field keep shooting up.
>
> You point out that the way to convince the field is to "demonstrate that the authors' insights into biology and cognition can be productively incorporated into computational ML approaches". We not only agree but in fact provide multiple such references in the paper, both in Section 2 and Section 4, especially.
>
> You also recommend that "[what] the authors need to make their argument convincing is specific examples of research done wrong and research done right". As mentioned, we provided examples of research "done right" (in our opinion). While there are many examples of "research done wrong", we disagree that pointing at them (especially by authors not yet well-established in the field) in a public paper is a desirable thing to do. We believe that it would be rather unethical. Instead, we focused on influential papers (and talks) by well-established scientists that have crucially shaped the field. The "boastful and unfortunate quotes from prominent figures" (adjectives by the reviewer) have arguably had a tremendous impact in defining research directions and terminology, and are included in the paper to illustrate the zigzagging and confusing trends and vocabulary. You seem to agree that these quotes have been unfortunate, so we do not understand how you see this as a weakness of our paper.
>
> You write in your review that "[t]he article reads as a stream-of-consciousness series of thoughts". By "stream-of-consciousness" (a term used in literary theory), do you mean that the article is poorly organised and confusingly written? If so, could you provide specific examples so that we can improve the manuscript?
>
> Finally, an important factor in the overall assessment of your review seems to be that our "thoughts" contain "many dated and incorrect, or at least arguable, points". However, it is not clear from the review which are these "many" incorrect points". We will next address the specific points mentioned in the review, which we believe we can discuss and solve confidently one by one:
>
> * Catastrophic forgetting: you see to have understood that we wrote the title and then changed our mind about the content. We have actually reviewed the manuscript carefully and this title has been accordingly carefully chosen. It is in fact a play on words, which we had expected it to be clear, since the title of the section is "catastrophic forgetting *of old concepts*", the text immediately after refers to a "classical result" (no free lunch theorem), and the topics of the paper are clearly detached from the technical notion of catastrophic forgetting. We are surprised that this was not clear, but the final version will have a foot note to further clarify it. As a matter of fact, the footnote was already written, including a mention to a prominent scientist who used it at a local presentation to refer to some of the problems we discussed in our paper. We omitted the foot note to avoid a potential disclosure of the identity, at least the affiliation of the authors. We hope that the misunderstanding is clarified and the reviewer reconsiders their position in this regard.
> * "The authors' take from the no-free-lunch theorem is incorrect": We humbly but strongly disagree with this statement. We agree with you that the no free lunch theorem motivates "choosing inductive bias appropriate for a task", this is in fact what this section is about. Supervision can be seen as a particular case of inductive bias, hence the mention of supervision in the discussion of the no free lunch theorem and inductive biases. We are willing to clarify this point to avoid confusion, but we disagree that we make an "incorrect" discussion of the classic no free-lunch theorem.
> * You also considers that our take on "implicit supervision" is incorrect. We humbly refer to  Locatello et al. (2018) (Best Paper Award at ICML 2019), which we cite and that makes a similar point, using exactly this term, "implicit supervision". Our take here is to review their relatively recent result and put it in perspective with the rest of the paper, old results and recent trends. We hope this clarifies this point.
> * Regarding the probabilistic perspective about supervised and unsupervised learning, we simply agree with you, as you in fact make the point we make in our paper: that supervised and unsupervised learning are not qualitatively different. We may disagree that this is something that every other researcher in the field would agree, judging for the literature, including textbooks (but this is not made explicit in the review). Thus, we do not see which is the "incorrect" part on our side, and in fact your specific concern.
>
> (_we briefly continue the response in the next comment_)

---

> > ### Author Response · Authors · 2021-08-10
> > **General response to the review (#2)**
> >
> > (_continues from previous comment_)
> >
> > * We also do not see what is the big concern in the point about supervised learning being more than classification. We mention this in one sentence of our paper that starts with "it should not be necessary to recall that", and is used to introduce the next point and for completeness. Again, we have no doubts that this point is clear to you, with 30-year experience in research and teaching, but we recall again that the field is composed by many other people who, as a matter of fact, do not have this entirely clear, as we can tell by our experience in the field too. In any case, this is just one sentence of the paper, not our claim of contribution. Is there anything incorrect? Is saying something at its worst true and well-known for everyone a reason for "clear rejection"?
> > * We also agree with you that machine learning is more than image classification. Where in the paper do we say the opposite? We also do not see what is incorrect about this. Granted, we do take image classification as the main source for our examples, because it is arguably one of the largest subfields in machine learning, and familiar to most researchers. But at no point are we denying the existence of other research directions.
> >
> > We hope our response addressed your concerns and we look forward to discussing further. Thank you again for the review.

---

> > ### Comment · Reviewer_pJTi · 2021-08-20
> > **appropriateness of work for NeurIPS**
> >
> > I appreciate the author's thoughtful and in-depth response. I read each of the authors' points and thank the authors for their diplomatic responses and elaborations to my critique. I am swayed by some of the small points the authors make in the bullets at the end of the rebuttal. However, like reviewer YQEE, I am not persuaded that the work is suitable for NeurIPS. All three of the reviewers have pointed out that the style and content of the work is that of a position paper or blog post or opinion piece. The authors themselves describe much of the paper as a 'review'. While this fact should not disqualify an acceptance at a top ML conference, this work is aimed at educating the audience and not offering deep, novel insights that have been investigated using experimental or theoretical tools of the field. The authors' rebuttal argues that while the paper may not offer an old fart like myself much novelty, it may have a message to an audience that is newer to the field or new to cognitive science/neuroscience. While true, I do not see a category of NeurIPS papers that are meant to review related fields or offer advice to newer researchers. I can see other venues at NeurIPS for this work, e.g., there is a NewInML workshop (https://vanyacohen.github.io/NewInML/), where I suspect the organizers would _welcome_ a contribution from folks such as yourselves, and you could contribute a Tutorial on cog sci/neurosci. The bottom line is that a NeurIPS paper should propose an idea that is novel to all researchers and to the entirety of the field. There is no shortage of researchers -- even at NeurIPS -- who already understand and appreciate the message you convey.

---

> > > ### Author Response · Authors · 2021-08-27
> > > **Type of paper, significance and contributions. Other concerns re: originality, quality and clarity?**
> > >
> > > Thank you for following up the discussion, we greatly appreciate it.
> > >
> > > We are pleased to learn that after reading our rebuttal you were "swayed" by some of our responses.
> > >
> > > If we understand it correctly, the remaining reason for your recommendation of rejection has to do with the _significance_ and the suitability for NeurIPS of our manuscript. Please let us briefly reply to the specific points you made in the last comment:
> > >
> > > > [T]he style and content of the work is that of a position paper or blog post or opinion piece.
> > >
> > > We fully agree that the style and content is that of a *position paper*. Admittedly, this is not _common_ at NeurIPS, but this type of manuscript is very common and valued in scientific venues more generally. Are position or opinion papers to be automatically rejected from at NeurIPS? We would love to read the official position from the Program or Area Chairs.
> > >
> > > _Blog post_ is less well defined than _position paper_. There are all kinds of blog posts, but we argue that our paper has substantially more scientific insight (review of the literature, connection of insights from different fields, etc.) than the vast majority of blog posts.
> > >
> > > > The authors themselves describe much of the paper as a 'review'. [...] [T]his work is aimed at educating the audience and not offering deep, novel insights that have been investigated using experimental or theoretical tools of the field. [...]
> > >
> > > To clarify, we indeed think that our paper has a strong _review_ component, but it is not a standard review of, for example, self-supervised learning methods. We argue that our paper provides 1) a critical review of recent publications trends in machine learning, 2) a review of biological learning from publications in cognitive science, neuroscience, developmental psychology, evolutionary biology, among other fields, and 3) a connection of the latter to machine learning research. In our opinion, many of the dots we connect in our paper are original and novel contributions, as we have argued both in the paper and the rebuttals. Therefore, our paper is not a mere educational piece aimed only for newcomers to the field of machine learning. It is a critical review with novel insights from interdisciplinary research. Quoting Reviewer 6UDa, "[t]he perspective advocated by the authors is provocative and original".
> > >
> > > > [A] NeurIPS paper should propose an idea that is novel to all researchers and to the entirety of the field.
> > >
> > > This is arguably not the case of every paper accepted at NeurIPS, and that is probably a good thing. As argued by many (see, for instance, [Slide 12 of the _Reviewer Slides for CVPR 21_](http://luthuli.cs.uiuc.edu/~daf/CVPR21TrainingMaterials/RefSlides.pdf), "many important things aren't all that novel" and "many novel things aren't all that important".
> > >
> > > ---
> > >
> > > To sum up, we have tried to convince you that the significance of our paper is higher than you have assessed, position which we obviously respect. Finally, besides _significance_ we would like to hear whether any concern remains regarding the other dimensions highlighted by NeurIPS in their [Reviewer Guidelines](https://neurips.cc/Conferences/2021/Reviewer-Guidelines), namely _originality_, _quality_ and _clarity_.
> > >
> > > Thanks again for your time invested in reviewing and discussing about our manuscript.

---

### Author Response · Authors · 2021-09-01
**Brief general comments**

We thank again all reviewers for their initial feedback and for further responding to our rebuttals.

We would like to note, as a general comment, that it seems the assessments of the paper have been greatly affected by the fact that this manuscript belongs to an unusual category at NeurIPS, that is position or opinion papers. Therefore, one of the main criticisms has been its suitability for the venue. Beyond that, the main criticism has been the _significance_. We would like to stress that, as a position paper, the contribution and significance of our paper greatly differs from the more standard papers that present empirical or theoretical results. Finally, we are glad to note that the reviewers have not highlighted other significant weakness, or mentioned aspects that remained unresolved after the rebuttal.

---

### Decision · Program_Chairs · 2021-09-27

**Decision:**

Reject

**Comment:**

This is a position paper, which is less common at NeurIPS.  It received very thorough reviews, and the authors responded in earnest.  However, the general consensus from the reviews was that the arguments posed by the paper did not have the depth or clarity required for a position paper to have a real impact as part of the NeurIPS program.

I would also encourage the authors to consider taking a less combative tone in future rebuttals.  Though the review process may seem adversarial, a lot of good can come out of it when it is approached as an opportunity for improvement.